# Influence of Viral Re-Infection on Head Kidney Transcriptome of Nervous Necrosis Virus-Resistant and -Susceptible European Sea Bass (*Dicentrarchus labrax,* L.)

**DOI:** 10.3390/v17020230

**Published:** 2025-02-07

**Authors:** Dimitra K. Toubanaki, Odysseas-Panagiotis Tzortzatos, Antonia Efstathiou, Vasileios Bakopoulos, Evdokia Karagouni

**Affiliations:** 1Immunology of Infection Group, Department of Microbiology, Hellenic Pasteur Institute, 11521 Athens, Greece; ptzortzatos@pasteur.gr (O.-P.T.); toniaef@pasteur.gr (A.E.); 2Department of Marine Sciences, School of The Environment, University of the Aegean, University Hill, Lesvos, 81100 Mytilene, Greece; v.bakopoulos@aegean.gr

**Keywords:** viral nervous necrosis, nervous necrosis virus, nodavirus, European sea bass, disease resistance, transcriptome, viral re-infection, host–pathogen interaction

## Abstract

Fish viral infections have great environmental and economic implications in aquaculture. Nervous necrosis virus (NNV) is a pathogen affecting more than 120 different species, causing high mortality and morbidity. Herein, we study how NNV re-infection affects the European sea bass (*Dicentrarchus labrax*, L.) head kidney transcriptome in disease-resistant and -susceptible sea bass families. To determine how each family responds to re-infection, we performed the RNA-sequencing analysis of experimentally NNV-infected *D. labrax*. Fish were experimentally infected in a long-term study, and one month after the last recorded death, all surviving fish were re-infected by the same NNV strain. Fish tissues were sampled 7 days upon re-infection. The transcriptome profiles of infected vs. non-infected fish revealed 103 differentially expressed genes (DEGs) for the resistant family and 336 DEGs for the susceptible family. Only a few pathways were commonly enriched in the two families, further indicating that the resistant and susceptible families utilize completely different mechanisms to fight the NNV re-infection. Protein–protein interaction analysis identified a variety of hub genes for the resistant and the susceptible families, quite distinct in their function on NNV resistance. In conclusion, NNV-resistant and -sensitive sea bass transcriptomes were analyzed following NNV survivors’ viral re-infection, offering a glimpse into how host attempts to control the infection depending on its genetic background in relation with virus resistance.

## 1. Introduction

Nervous necrosis virus (NNV), or nodavirus, is the causative agent of viral encephalopathy and retinopathy (VER) disease, which can cause high mortality and morbidity in more than 120 different species from marine and freshwater environments, and it is responsible for great economic losses in the aquaculture industry, affecting human nutrition and the environment [1,2,3,4]. NNV belongs to the genus Betanodavirus and is a member of the Nodaviridae family, characterized by single-stranded RNA genome with two positive-sense molecules (RNA1 and RNA2) in a non-enveloped icosahedral capsid. Betanodaviruses can be classified into four different genotypes: red-spotted grouper nervous necrosis virus (RGNNV), striped jack nervous necrosis virus (SJNNV), tiger puffer nervous necrosis virus (TPNNV), and barfin flounder nervous necrosis virus (BFNNV), based on the T4 region of RNA2 [5]. The disease is characterized by severe damage of the fish nervous system (e.g., brain, retina, spinal cord), and the clinical symptoms include abnormal swimming behavior, loss of appetite, swim bladder hyperinflation, coloration abnormalities, eventually resulting in the potential death of infected hosts [2,3,6].

European sea bass (*Dicentrarchus labrax*) is one of the most susceptible hosts of NNV. It is a teleost fish found in the Mediterranean, north-eastern Atlantic Ocean, and Black Sea. It is well characterized [7] and economically important, since it is the main marine fish farmed in the Mediterranean basin, along with gilthead seabream [8,9], which is considered an asymptomatic carrier of NNV [10]. In general, the onset of a disease outbreak depends both on pathogen virulence and the host immune response [11], and vaccination may be used to protect fish from NNV [12]. However, the use of commercial vaccines to protect sea bass against VNN in the Mediterranean is still limited due to the cost, technical, and logistic problems presented when combined with other vaccines [6,13]. As an alternative solution, selective breeding in order to have traits such as disease resistance has gained attention in recent years [14].

The molecular bases of disease resistance mechanisms are often complex, and resistance is rarely attributable to a single immune function characteristic [15]. The development of fish genetic maps and -omics technologies allowed for the identification of several genes and pathways which are related to host resistance against several pathogens in a range of fish hosts [15,16,17,18,19,20,21]. For example, disease-resistant fish seem to have significantly higher expressions of pro-inflammatory genes and transcription factors, like those in the salmon head kidney following viral infection [18]. Resistant fish appear to have a limited and prolonged immune response to the virus, while susceptible fish have an acute short response to viral infection [19]. In another study, genes related to cytokine activity and inflammatory response were up-regulated in susceptible fish, but that mechanism failed to protect the salmon against the virus, whereas the resistant fish had a milder immune response, including the up-regulation of genes relating to the M2 macrophage system, which seems to be a more effective way of surviving a viral infection [20]. A transcriptome profile comparison of Japanese flounder both resistant to and susceptible against bacterial infection showed that genes related to hematopoietic cell lineage, innate immune-related inflammatory factors, antigen processing and presentation, and T/B cell receptor signaling pathways were significantly enriched in the resistant family. These results indicate that the resistance molecular mechanism is controlled by multiple genes in the immune signaling pathway [15]. In *Aeromonas salmonicida* bacterium-resistant/susceptible turbot, a transcriptome analysis revealed that the resistant family displayed a more controlled inflammatory response to infection, and several up-regulated genes in the head kidney were related to antigen presentation and T-cell activity [21].

The genetic assessments of European sea bass resistance to NNV infection suggests moderate heritability for resistance [13,22,23]. A strong quantitative trait locus (QTL) at the LG12 chromosome was reported [13,23], along with putative traits at LG8, LG15, and LG19. Significant SNPs, associated with the putative QTL genotype at LG12, included ITPK1 (inositol tetrakisphosphate 1-kinase 1), PLK4 (polo-like kinase 4), REEP1 (receptor expression-enhancing protein 1), CHMP2 (charged multivesicular body protein 2), MRPL35 (mitochondrial ribosomal protein L35), and SCUBE1 (signal peptide-CUB-EGF (epidermal growth factor) domain-containing protein 1). The highest significant SNP was located within the intron of the HSPA4L gene (heat shock protein family A, member four) which belongs to the heat shock proteins (HSPs) family. HSPs are implicated in cell responses to harmful circumstances and protect cells from stress [13]. Genetic analysis provides valuable insights for selective breeding. However, in order to understand host resistance mechanisms, more parameters should be analyzed, including transcriptome, proteome, and metabolic profile studies of resistant and susceptible fish.

In the present study, in order to investigate the VNN resistance mechanism of European sea bass, two genetically distinct families were chosen for head kidney transcriptome analysis; i.e., one NNV-resistant and one susceptible family originated from a well-studied family-based breeding program with 100 sea bass families [13]. Transcriptome analysis was performed to compare NNV-resistant and NNV-susceptible families’ responses to viral re-infection in the fish head kidney because it is one of the most important lymphoid tissues in teleosts and plays a crucial role in the immune responses following an infection [24]. Based on the transcriptome data, the resistant and susceptible families’ responses were assessed in three complementary ways: gene ontology (GO) enrichment analysis, pathway (KEGG and reactome) analysis, and protein–protein interaction (PPI) analysis, which were used to reveal the key genes and pathways important for disease resistance. The obtained data are critical to understanding the functional basis of genetic resistance to NNV in European sea bass. Overall, such studies could provide general perspectives on host resistance mechanisms against viral pathogens in teleosts, with implications on rational prognostic and therapeutics strategies for sustainable aquaculture.

## 2. Materials and Methods

### 2.1. Experimental Fish

The present experimentation was performed in the frame of a larger project involving two families of European seabass with different levels of resistance to NNV infection (i.e., an NNV-resistant family (R) and an NNV-susceptible family (S)), as determined on a family-based breeding program [13]. Briefly, healthy non-vaccinated fish were used and transported by Nireus Aquaculture S.A. at the Laboratory of Ichthyology-Aquaculture and Aquatic Animal Health (ICHTHYAI), Department of Marine Sciences, University of The Aegean, Greece. A total of 300 randomly selected fish from both families (weight: 125.85 ± 29.8 g) were acclimatized for 3–4 days in 1 m^3^ cylindroconical fiberglass tanks connected to a closed recirculated sea water system, with a 20 m^3^ total volume capacity. Water was recirculated via a 14 m^3^ h^−1^ sea water pump, filtered via a sand filter, disinfected via 5 × 39 W UV-C lamps, treated in a biological filter, and aerated via air stones connected to five 150 L h^−1^ air pumps. Seawater in the system was renewed by 1/3 every 1.5 months. The seawater temperature was maintained at 22–23 °C during the acclimatization period. Salinity was 3.8–3.9%, dissolved O_2_ was maintained above 4.8 mg L^−1^, total ammonia nitrogen and nitrite was kept below 0.05 ppm and 0.5 ppm, respectively, and nitrate levels were maintained below 40 mg L^−1^. pH ranged between 7.9–8.1. Temperature, dissolved oxygen, and nitrogen metabolites were measured daily, while salinity and pH were measured on a weekly basis. Fish were reared in a 12 h light–12 h dark photoperiod and were fed with 1–2% of their biomass commercial (Feedus, Blueline) diet 3 times a day with 6 h intervals.

### 2.2. Nervous Necrosis Virus Challenge

NNV was originally isolated from naturally infected *D. labrax* (genotype: RGNNV [25]) and was propagated as previously described [26]. Briefly, fish brains were homogenized in EMEM (Eagle Minimum Essential Medium; Sigma-Aldrich, Steinheim, Germany) or Leibovitz L15c medium (Biochrom, Berlin, Germany) containing 10% FBS (Fetal bovine serum; Biochrom, Berlin, Germany). Following inoculation, the SSN-1 cells (The European Collection of Animal Cell Cultures, Salisbury, UK) were grown at 26 °C in Falcon Primaria cell culture flasks (Becton Dickinson Labware, Franklin Lakes, NJ, USA) containing Leibovitz’s L15 medium, supplemented with 10% FBS, 100 u mL^−1^ penicillin, 100 μg mL^−1^ streptomycin, and 2 mM glutamine (Gibco, Paisley, UK). Virus titration was performed on monolayers of SSN-1 cells grown in a 96-well plate. Viral suspensions were prepared with 12-fold serial dilutions in EMEM supplemented with 10% FBS. Quadruplicates of 50 µL of each dilution were added in a 96-well plate seeded with SSN-1 cells. Cultures were incubated at 26 °C for 6 days. During this period, the cell monolayers were observed for the appearance of a cytopathic effect (CPE), and the final titer, expressed as TCID_50_ mL^−1^, was estimated by the end-point titration method [27].

A summary of the experimental setup is given in Figure 1. Prior to infection, groups of 20 and 50 fish were transported to separate tanks (in triplicates), for family R and family S, respectively; the temperature was gradually raised to 27.0–27.2 °C and the fish were acclimatized for 7 more days. The temperature remained constant to 27.0–27.2 °C throughout the experiment. Before the infection experiment, total RNA was extracted from three randomly selected individuals’ brains and amplified with an NNV RT-qPCR assay using the Quantitect Probe RT-PCR master mix (Qiagen) to ensure that those specimens were not infected. Sea bass were challenged by intramuscular injection in the dorsal muscle with 7 × 10^6^ TCID_50_ mL^−1^ of nodavirus-containing supernatant (200 µL; serially diluted with PBS from a 1 × 10^11^ TCID_50_ mL^−1^ nodavirus-containing supernatant stock). As the negative control group, uninfected sea bass from the same families were mock-challenged with 100 µL PBS. Fish were monitored twice a day, and mortalities were recorded up to the end of the experimentation. The first stage of experimentation was up to 28 dpi and was previously described in [28]. After an additional period of 2 weeks (42 days post infection in total), the fish that had survived were re-infected by intramuscular injection in the dorsal muscle with 1 × 10^5^ TCID_50_ mL^−1^ of nodavirus-containing supernatant (100 µL). All deceased fish were removed, and one week following re-infection, 9 fish of the remaining live population (3 samples from each tank) were randomly selected. Before sampling, the specimens were anesthetized with 0.2% phenoxyethanol and weighed. Following, head kidney and brain tissues were removed aseptically and were either subjected in RNA isolation immediately or stored in RNAlater (Qiagen, Hilden, Germany) at −80 °C. The same procedure was followed for 9 fish belonging to the negative control groups for each family. For the determination of the brain viral load, a quantitative RT-qPCR assay was used (*n* = 5) [29].

### 2.3. Ethics Statement

Experimentation was performed in the Laboratory of Ichthyology-Aquaculture and Aquatic Animal Health (ICHTHYAI) (Government Issue 1255/28-4-2016). ICHTHYAI has been granted all the required permits for producing (EL 83 BIObr 01), supplying (EL 83 BIOsup 01), and experimenting on aquatic organisms (EL83 BioExp 01), according to the Presidential Decree 56/2013 conforming to Directive 2010/63/ΕΕ (Decision No 4053/14-3-2017 of the competent Regional Veterinary Authority). Fish were euthanized using a procedure listed on the appropriate license, and the protocol for the experimental infection performed in this study has been approved by Decision No 5379/4-4-2017 of the competent Regional Veterinary Authority.

### 2.4. RNA Extraction

Total RNA from the head kidney was extracted using a Trizol-based protocol to perform total RNA sequencing. Tissues (~75 mg) from NNV-infected (*n* = 9) and non-challenged (*n* = 5) fish were homogenized using the TissueLyzer mechanical homogenizer with 5 mm steel beads (Qiagen). The TRIzol^®^ Reagent solution (Invitrogen, Carlsbad, CA, USA) was used for total RNA extraction, following the manufacturer’s instructions. RNA quantity and quality were assessed by spectrophotometry (NanoDrop 2000), a Qubit 2.0 fluorometer (Thermo Fisher Scientific, Waltham, MA, USA) and the RNA integrity number (RIN) was measured using a 2100 Bioanalyzer instrument (Agilent Technologies, Santa Clara, CA, USA). Only RNA samples of high quality (RIN ≥ 7) were used for constructing the cDNA libraries, and each samples’ transcriptome was analyzed individually.

### 2.5. cDNA Library Construction, Sequencing, and Transcriptome Mapping

RNA samples were used to create the sequencing libraries by using the Ion Total RNA-Seq Kit v2 kit (Thermo Fisher Scientific, Waltham, MA, USA). The analysis was comprised of 16 individual samples: 2 genotypes (resistant, susceptible) × 2 challenge states (challenged, control) × 3–5 biological replicates. The 16 libraries were prepared according to the manufacturer’s instructions and then sequenced using Ion Torrent technology (Ion S5XL platform, (Thermo Fisher Scientific Inc., Waltham, MA, USA)). Raw data were processed to remove adapters and were normalized for inherent systematic or experimental biases using the Bioconductor package DESeq2, v. 1.46.0. The reads were aligned to the *Dicentrarchus labrax* (seabass_V1.0-GCA_000689215.1) reference genome [7] with hisat2 (v. 2.2.1) and bowtie2 (v. 2.4.5). Post-mapping quality control was assessed with the Bioconductor package metaseqR2, v. 1.18.0 [28,29]. All raw data reads are available in the NCBI database (Accession No. PRJNA1030357).

### 2.6. Differential Expression and Enrichment Analyses

Basic differential expression analysis was performed with the Bioconductor packages metaseqR2 and/or edgeR [30,31,32]. A transcript was considered as a differentially expressed transcript if the adjusted *p*-value (FDR) threshold was less than 0.05 (significance level) and the log2-transformed fold change (log2FC) was more than 1.5. The function of the identified differentially expressed transcripts was analyzed in OmicsBox software (Version 3.2.4) by first using BLASTX against an NCBI non-redundant (NR) database to search for the possible top hit proteins (accessed on 6 June 2024). To obtain high-quality results, the ‘*Actinopterygii*’ (Taxid:7898) taxonomy filter was applied. Thereafter, blasted sequences were subjected to gene ontology (GO) mapping and annotation with default parameters. GO functional enrichment and pathway analysis were carried out by BLAST2GO (BioBam Bioinformatics, Valencia, Spain) using the total transcripts dataset as the reference background. The annotated DEGs were subjected to Fisher’s exact test and were considered significantly enriched in GO terms when their Bonferroni adjusted *p*-value was less than 0.05. The results were reduced to the most specific terms. Enriched KEGG pathways were determined by Gene Set Enrichment Analysis (GSEA), with adjusted *p*-value less than 0.05. All bubble plots and pathway summaries were generated using the SRplot [33].

### 2.7. Construction and Analysis of PPI Networks and Functional Annotation

To further investigate the relationship of resistance-related genes following virus re-infection, the Search Tool for the Retrieval of Interacting Genes/Proteins database (STRING v11.5; http://www.string-db.org, accessed on 27 November 2024) was used to construct their PPI network [34]. All blasted differentially expressed transcripts were manually curated and translated to reference gene names orthologous to the model organism, zebrafish (*Danio rerio*; Taxid: 7955), in the Uniprot database (accessed on 27 November 2024) [https://www.uniprot.org/].

### 2.8. Real-Time Quantitative Polymerase Chain Reaction (qPCR) Validation

Real-time PCR assays were carried out to confirm the differential expression data of transcriptome analysis. Specific primers used for the expression analysis of genes from both families (Appendix A) were designed utilizing Primer-Blast (https://www.ncbi.nlm.nih.gov/tools/primer-blast/, accessed on 12 July 2024), IDT_Primer Quest (https://www.idtdna.com/pages/tools/primerquest, accessed on 12 July 2024), and Primer3Plus (https://www.primer3plus.com/, accessed on 12 July 2024). All qRT-PCR primers were designed according to the minimum information needed for the publication of qRT-PCR experiment (MIQE) guidelines [35]. The potential primer secondary structures (homo- or cross-dimers and hairpin structures) and primer specificity were checked with IDT_Primer Quest and Primer-Blast, respectively. The elongation factor gene was chosen in our study as the reference gene. Before sample quantification experiments, the specificity of each primer pair was studied using positive and negative samples.

Total RNA was used as a template to synthesize cDNA using the QuantiNova Reverse Transcription kit (Qiagen) according to the manufacturer’s instructions. Approximately 5 μg RNA were used as the input material. Real-time PCR reactions were performed with QuantiNova SYBR Green PCR (Qiagen) using 1 µL of a 1:10 dilution of cDNA. Primers for all genes were used at 500 nM. The thermal conditions used were as follows: 2 min at 95 °C of pre-incubation followed by 40 cycles at 95 °C for 10 s and 60 °C for 30 s. An additional temperature ramping step was utilized to produce melting curves from 62 to 95 °C to verify the amplification of a unique single product on all samples. All reactions were performed in technical triplicate using a RotorGene Q PCR Detection System (Qiagen). The quantification was performed according to the comparative CT method [36]. The value for each experimental condition was expressed as normalized relative expression, calculated in relation to the values of the control group and normalized against those of the reference gene (by its geometric average).

## 3. Results

### 3.1. Experimental Challenge of NNV-Resistant and -Susceptible D. Labrax Families

The experimental NNV challenging of the *D. labrax* fish of both resistant (R) and susceptible (S) families was utilized to study the NNV infection progress, as depicted in Figure 1.

Following the first injection challenge, in NNV-resistant fish (R), typical signs of VNN were observed from day 3 and the first death occurred on day 4, while the NNV-susceptible fish (S) had apparent VNN symptoms and the first death occurred on day 2. The R family appeared to experience a decline in the survival of infected fish from day 4 to day 15 post infection (86.3% survival/ 13.7% mortality), which ceased until the end of the experiment. On the contrary, the S family had a sharp decline in fish survival from day 2 to day 11 post infection, with a maximum mortality on day 5 (50.5% survival/ 49.5% mortality) (Figure 2). No mortalities were recorded following the re-infection of both families. No mortalities were registered in non-infected sea bass, whose survival remained 100% up until the end of the experiment.

The sea bass brain was tested for viral NNV RNA1 presence in both families at the final time point, i.e., 7 days post re-infection (dpri). The viral load of the surviving fish 7 dpri was 20.1 ± 1.6 × 10^11^ TCID_50_ of the NNV per µg of total RNA for the R family and 26.0 ± 8.9 × 10^11^ TCID_50_ of the NNV per µg of total RNA for the S family. Despite the presence of the virus in the brain, no phenotypic or behavioral changes were observed in infected fish and no natural fish loss (i.e., not related to NNV infection) occurred during the experiment following re-infection. No viral load was observed in non-infected groups.

### 3.2. Summary and Assessment of NNV-Challenged D. Labrax Head Kidney RNA Sequencing (RNA-Seq) Data

Head kidney total RNA isolated from both resistant and susceptible NNV-infected (*n* = 10) and non-infected (*n* = 6) fish were subjected to RNA-seq. The sequencing of the 16 libraries yielded a total of 224,837,867 and 123,140,003 sequences for NNV-infected and non-infected head kidney samples, respectively. Raw data reads are available in the NCBI database (Accession No. PRJNA1030357).

### 3.3. Differential Expression Analysis Between Infected and Non-Infected Groups

Differentially expressed transcripts were identified according to log_2_FC ≥ 1.5, and the adjusted *p*-value (FDR) threshold was set as less than 0.05. The transcriptome data were compared among the four groups: the resistant infected fish (RI), the susceptible infected fish (SI), the resistant non-infected fish (RNI), and the susceptible non-infected fish (SNI). The pairwise comparison of RI vs. RNI identified 57 significantly up-regulated genes and 46 significantly down-regulated genes, while the SI vs. SNI comparison resulted in 219 significantly up-regulated genes and 117 significantly down-regulated genes. The pairwise comparisons were also performed between the resistant and susceptible families; the RI vs. SI comparison resulted in 215 significantly up-regulated genes and 116 significantly down-regulated genes. The RNI vs. SNI comparison identified 57 significantly up-regulated genes and 62 significantly down-regulated genes.

The up-regulated and down-regulated DEGs corresponding to the described comparisons are depicted in Figure 3A. Most DEGs were overexpressed after infection for both families (RI vs. RNI and SI vs. SNI comparisons). Also, more up-regulated DEGs were found in the RI vs. SI comparison. The number of common de-regulated DEGs between RI vs. RNI and SI vs. SNI was relatively low (20 common genes), reflecting a totally different head kidney response upon re-infection between the resistant and the susceptible family (Figure 3B). A differential gene expression clustering analysis indicated high intragroup similarity and intergroup differences between all compared groups (Figure 3C and Appendix A).

### 3.4. Resistant Infected (RI) Versus Resistant Non-Infected NNV (RNI) Experimental Group Analysis Results

#### 3.4.1. GO Enrichment Analysis of DEGs Between Resistant Infected (RI) and Non-Infected NNV (RNI) Experimental Groups

In order to identify the potential molecular mechanisms needed to overcome NNV infection in the resistant family, the functional enrichment analyses were performed between the resistant infected (RI) and non-infected NNV (RNI) experimental groups. For the up-regulated DEGs, the GOs result from the enrichment analysis (adj. *p* < 0.05) are listed in Appendix A and the top 30 GOs are shown in Figure 4A. The most enriched category was the biological process (25/43), followed by the cellular component (9/43) and the molecular function (9/43) categories. For the down-regulated DEGs, the GOs resulting from enrichment analysis are listed in Appendix A and the top 30 GOs are shown in Figure 4B. The most enriched category was the biological process (25/47), followed by the cellular component (11/47) and the molecular function (11/4) categories.

Among the top enriched categories for both the up- and down-regulated genes were (1) signaling, immune system process, and vesicle-mediated transport; (2) hydrolase activity, molecular transducer activity, and transferase activity; (3) cytosol, plasma membranes, and mitochondrion. The categories uniquely enriched in the up-regulated genes were cytoplasmic vesicle, structural molecule activity, mitochondrion organization, protein catabolic process, catalytic activity acting on a protein, carbohydrate metabolic process, amino acid metabolic process, the generation of precursor metabolites and energy, nucleoplasm, nucleobase containing small molecule metabolic process, molecular function regulator activity, chromatin organization, nucleocytoplasmic transport, autophagy, DNA metabolic process, and cytoskeletal protein binding. The categories uniquely enriched in the down-regulated genes were transmembrane transport, transporter activity, cell adhesion, endosome, catalytic activity acting on a protein, nucleus, GTP binding, catalytic activity acting on DNA, receptor ligand activity, circulatory system process, nervous system process, extracellular matrix organization, muscle system process, endocrine process, cell junction organization, inflammatory response, Golgi apparatus, vitamin metabolic process, extracellular matrix, and lipid binding.

#### 3.4.2. Pathway Analysis of DEGs Between Resistant Infected (RI) and Non-Infected NNV (RNI) Experimental Groups

Pathway analysis was applied between the resistant infected (RI) and non-infected NNV (RNI) experimental groups. All annotated transcripts were run against the KEGG and Reactome databases. It was found that 97 pathways were enriched in the KEGG database, while 19 pathways were enriched in the Reactome database. The enriched pathway categories for both databases are summarized in Table 1 and analyzed in Appendix A and the top 20 pathways are shown in Figure 5. These pathways are involved in (1) immune system: antigen processing and presentation, class I MHC-mediated antigen processing and presentation, NOD-like receptor signaling, C-type lectin receptor signaling, cytosolic DNA-sensing, neutrophil extracellular trap formation, and IL-17 signaling; (2) signaling molecules and interaction: cytokine–cytokine receptor interaction, viral protein interaction with cytokine and cytokine receptors, and cell adhesion molecules; (3) signal transduction: Rap1 signaling, HIF-1 signaling, Ras signaling, signaling by PTK6, and RHO GTPase cycle; (4) cell growth and death: apoptosis, necroptosis, p53 signaling, and the apoptotic cleavage of cellular and cell adhesion proteins; (5) genetic information processing: proteasome, RNA degradation, RNA polymerase, mRNA surveillance, and spliceosome; (6) metabolism: amino acid metabolism, carbohydrate metabolism, oxidative phosphorylation, and fatty acid degradation; and (7) viral infectious diseases, immune diseases, and neurodegenerative diseases.

#### 3.4.3. Protein-Protein Interaction (PPI) Analysis of DEGs Between Resistant Infected (RI) and Non-Infected NNV (RNI) Experimental Groups

To gain further insights into the disease resistance mechanisms, the zebrafish orthologs of all differentially expressed genes in between the resistant infected (RI) and non-infected NNV (RNI) experimental groups were manually determined. Of the 140 proteins, 105 were found in the Search Tool for the Retrieval of Interacting Genes/Proteins database of the STRING database and analyzed in terms of gene clustering (Figure 6). PPI analysis with k-means clustering resulted in five distinct clusters containing more than three genes. The hub genes include vegfc (vascular endothelial growth factor C); aldh18a1 (delta-1-pyrroline-5-carboxylate synthetase); canx (calnexin/ MHC class I antigen, partial); mxf (interferon-induced GTP-binding protein Mx) and alox12 (arachidonate 12-lipoxygenase); and ddit4 (DNA damage-inducible transcript 4 protein).

### 3.5. Susceptible Infected (SI) Versus Susceptible Non-Infected NNV (SNI) Experimental Groups Analysis Results

#### 3.5.1. GO Enrichment Analysis of DEGs Between Susceptible Infected (SI) and Non-Infected NNV (SNI) Experimental Groups

In order to identify potential molecular mechanisms to overcome NNV infection by the susceptible family, the functional enrichment analyses were performed between the susceptible infected (SI) and non-infected NNV (SNI) experimental groups. For the up-regulated DEGs, the GOs resulting from the enrichment analysis (adj. *p* < 0.05) are listed in Appendix A and are shown in Figure 7A. The most enriched category was molecular function (8/16), followed by the biological process (6/16) and the cellular component (2/16) categories. For the down-regulated DEGs, the GOs resulting from the enrichment analysis are listed in Appendix A and are shown in Figure 7B. Only four GO categories were enriched belonging to the cellular component (2/4), the biological process (1/4), and the molecular function (1/4) categories.

The categories uniquely enriched in the up-regulated genes are signaling, the regulation of DNA-templated transcription, transcription regulator activity, DNA binding, GTP binding, molecular transducer activity, cell motility, intracellular membrane-bounded organelle, hydrolase activity, anatomical structure development, GTPase activity, immune system process, molecular function regulator activity, cell differentiation, and receptor ligand activity. The categories uniquely enriched in the down-regulated genes are cellular process, plasma membrane, and transporter activity. The only category enriched for both up- and down-regulated genes was the extracellular space.

#### 3.5.2. Pathway Analysis of DEGs Between Susceptible Infected (SI) and Non-Infected NNV (SNI) Experimental Groups

Pathway analysis was applied between the susceptible infected (SI) and non-infected NNV (SNI) experimental groups. All annotated transcripts were run against the KEGG and Reactome databases. It was found that 36 pathways were enriched in the KEGG database, while 35 pathways were enriched on the Reactome database. The enriched pathway categories for both databases are summarized in Table 2 and analyzed in Appendix A, and the top 20 pathways are shown in Figure 8. These pathways are involved in (1) the immune system: NOD-like receptor signaling, C-type lectin receptor signaling, cytosolic DNA-sensing, T cell receptor signaling, IL-17 signaling, cytokine signaling in immune system, toll-like receptor cascades, MyD88-dependent cascades, and the innate immune system; (2) signaling molecules and interaction: interleukin-1 family signaling and the regulation of NF-kappa B signaling; (3) signal transduction: NRIF signals cell death from the nucleus; (4) cell growth and death: cell cycle, necroptosis, and the regulation of actin cytoskeleton; (5) genetic information processing: proteasome, RNA polymerase, and nucleotide excision repair; (6) metabolism: carbohydrate metabolism, oxidative phosphorylation, and nicotinate and nicotinamide metabolism; (7) viral infectious diseases and immune diseases, but no neurodegenerative diseases.

#### 3.5.3. Protein–Protein Interaction Analysis of DEGs Between Susceptible Infected (SI) and Non-Infected NNV (SNI) Experimental Groups

The zebrafish orthologs of all differentially expressed genes in between the susceptible infected (SI) and non-infected NNV (SNI) experimental groups were manually determined. Of the 321 proteins, 217 were found in the Search Tool for the Retrieval of Interacting Genes/Proteins database of the STRING database and analyzed in terms of gene clustering (Figure 9). PPI analysis with k-means clustering, resulted in four distinct clusters containing more than three genes. The hub genes include egr1 (early growth response protein 1), jun (jun proto-oncogene, AP-1 transcription factor subunit), fosb (FBJ murine osteosarcoma viral oncogene homolog B); mmp13b and mmp19 (matrix metallopeptidases); and galr1b (galanin receptor 1b), a member of the G-protein-coupled receptor 1 family.

### 3.6. qRT-PCR Assay of Selected Genes in the Head Kidney of NNV-Resistant and NNV-Susceptible D. labrax Families

The expression levels of six selected genes were analyzed by quantitative RT-PCR to validate the DEGs identified by RNA-seq. The analyzed genes were the arylsulfatase g (arsg), g-protein-coupled receptor 84 (GPR84), and transmembrane protein 120a (TACAN), which were highly expressed by the resistant family, and zinc finger bed domain-containing protein 1 (SUMO) and hydroxycarboxylic acid receptor 2-like (12S-HETE) and collagenase 3-like (mmp13), which were highly expressed by the susceptible family. As shown in Appendix A, the expression of the genes for both families when the infected fish were compared with non-infected fish were consistent with the expression results obtained from the transcriptome analysis for each family.

## 4. Discussion

European sea bass is an economically important marine fish species widely farmed in the Mediterranean, and nervous necrosis virus is causing significant economic losses to the aquaculture industry [3]. It has been reported that betanodavirus survivors from natural infections are resistant to the disease recurrence [37], suggesting a protective immune response of the fish to infection. In another study, it has been reported that the surviving fish were found to produce neutralizing antibodies, which possibly explains their resistance to natural re-infection [6]; however, in a recent study on convalescent sevenband groupers, almost no NNV-neutralizing antibodies were detected, even though they were strongly protected against re-infection with NNV [38]. Thus, the potential molecular mechanisms of immune responses against NNV, especially following survivors’ re-infection, remains poorly understood.

The viral load in resistant sea bass brain was found to be slightly lower than the susceptible sea bass brain; however, both families had a considerable amount of the replicated virus in the target organ. This observation is in agreement with previous studies [20], confirming that fish with high levels of genetic resistance to the virus are actively infected; thus, genetic resistance cannot be entirely due to the inability of the virus to infect the fish. Although VNN infection is primarily localized to the brain, systemic immune responses are anticipated to manifest in the head kidney. To this end, numerous differentially expressed genes (DEGs) were identified in the head kidneys of the resistant infected vs. non-infected (RI vs. RNI) and the susceptible infected vs. non-infected (SI vs. SNI) experimental groups. The resistant and susceptible families’ responses were studied utilizing different approaches, i.e., gene ontology (GO) enrichment, pathway (KEGG and Reactome) enrichment, and protein–protein interaction (PPI) analysis, in order to examine the expression patterns and obtain a more comprehensive understanding of the biological functions of DEGs from each family. The ‘uniquely’ implicated components of each family response resulting from these analyses are summarized in Table 3.

The DEGs between RI vs. RNI were significantly enriched in genetic information processing, organismal systems (including immune system), and viral infection-related pathways, while the DEGs between SI vs. SNI were significantly enriched in immune-related and molecular function activity pathways. The GO terms ‘immune system process’ and ‘signaling’ were enriched in the up-regulated transcripts of both families, as expected, but we also found enrichment in the down-regulated transcripts of the resistant family.

In the up-regulated genes of the resistant family, many enriched GO terms were related to metabolism (e.g., ‘carbohydrate metabolic process’, ‘amino acid metabolic process’, ‘generation of precursor metabolites and energy’, and ‘nucleobase containing small molecule metabolic process’), which may be attributed to the host’s attempt to overcome lower energy availability as an effect of appetite loss, as described for other fish viruses [20]. Other metabolism-related GO terms were enriched in the down-regulated transcripts of the resistant family, including ‘vitamin metabolic process’, ‘lipid metabolic process’, and ‘lipid binding’. In vitro studies of RGNNV infection mechanisms proved that the virus requires lipogenesis for infection and induced the formation of lipid droplets [39]; therefore, the down-regulation of the related genes may reflect an attempt of the resistant sea bass to control the infection.

In addition, GO terms related to ‘nervous system process’ and ‘muscle system process’ were found to be down-regulated in the resistant family, which possibly makes an effort to control the disease symptoms (i.e., nervous system damage and abnormal swimming).

‘Vesicle-mediated transport’ and ‘cytoplasmic vesicle’ terms were also enriched exclusively in the resistant family, implying a crucial role of vesicle-mediated transport in response to the NNV infection. It has been suggested that the betanodavirus enters host cells mainly via clathrin-mediated endocytosis in a cholesterol-, pH-, and cytoskeleton-dependent manner, involving clathrin-coated vesicles containing internalized viruses [40]. In general, viruses hijack such cellular pathways to promote their propagation; however, the vesicle-related term enrichment in the resistant family may indicate the fish’s effort to modify its cells normal vesicular transport pathways to defend itself from the infection. For example, the cells could recognize and degrade viral components either via pattern recognition receptor detection, compartmentalization, and phagocytosis stimulation or by aggresome formation and autophagy induction [41]. On the other hand, the susceptible sea bass attempts to control infection utilizing specific molecules’ activity by enriching genes associated more with molecular function terms. The ‘transcription regulator activity’ is highly enriched in the up-regulated transcripts, since viral transcription regulators of gene expression are central to disease pathogenesis due to their ability to control the expression of both viral and host genes [42]; therefore, sea bass tries to effectively control their activity.

The pathway enrichment analysis was performed using both the KEGG and Reactome pathway databases. It was found that some immune-related pathways (‘NOD-like receptor signaling’, ‘C-type lectin receptor signaling’, ‘cytosolic DNA sensing’, and ‘IL-17 signaling’) were enriched in both resistant and susceptible fish, while different immune system components seem to be involved in each family response. The resistant family utilizes pathways related to ‘class I MHC-mediated antigen processing and presentation’, ‘neutrophil extracellular trap formation’, ‘viral protein interaction with cytokine and cytokine receptor’, as well as ‘cytokine–cytokine receptor interaction’ and ‘cell adhesion molecules’. The ‘cytokine–cytokine receptor interaction’ pathway was also found to be enriched in Atlantic salmon fry challenged with the infectious pancreatic necrosis virus in resistant fish at a late time point (20 dpi) [20]. The response in the susceptible family depends mostly on ‘T cell receptor signaling’, ‘cytokine signaling in immune system’, ‘toll-like receptor cascades’, ‘MyD88-dependent cascades’, the ‘innate immune system’, ‘interleukin-1 family signaling’, and the ‘regulation of NF-kappa B signaling’.

Apoptosis related-pathways (i.e., ‘apoptosis’, ‘p53 signaling’, and ‘apoptotic cleavage of cellular and cell adhesion proteins’) are enriched only in the resistant family, while the ‘necroptosis’ pathway is enriched by both families. When an organism is infected, its cells can regulate (or “program”) their death to tailor immune responses, thereby changing the impact that their death will have on the surroundings [43]. Necroptosis is a non-apoptotic form of cell death which has evolved to detect pathogens and promote tissue repair. The process culminates with the loss of membrane integrity and passive leakage of intracellular contents (e.g., cytokines, DAMPs, and PAMPs), which provide pro-inflammatory cues that recruit immune cells but also induces inflammation, which can be detrimental to the host. Apoptosis induces the breakdown of the nuclear membrane, the cleavage of many intracellular proteins, membrane blebbing, the breakdown of genomic DNA into nucleosomal structures, and the release of cytochrome c from mitochondria [43]. Apoptotic death usually leads to immunologically silent responses, and its activation does not promote a significant inflammatory response, thereby preserving homeostatic integrity [44]. Its utilization by the resistant family in combination with necroptosis, in a balanced way, is possibly responsible for better outcomes of host health.

Generic viral disease (‘Influenza A’ and ‘Coronavirus disease—COVID-19’) pathways were enriched in both families, in accordance with similar studies [20], as well as ‘RNA polymerase’ and ‘proteasome’ in the genetic information processing KEGG category. However, the ‘RNA degradation’, ‘mRNA surveillance’, and ‘spliceosome’ pathways were only enriched in the resistant family, indicating a more intense effort to control the host cell machinery response to the RNA virus.

Protein–protein interaction analysis of DEGs from each family resulted in discrete clustering with a variety of hub genes. Interactions of the resistant family are linked by vegfc, aldh18a1, canx, mxf, alox12, and ddit4. The **vascular endothelial growth factor (VEGF) C** (vegfc) is a member of the VEGF family, which are secreted polypeptides acting through a family of cognate receptor kinases in endothelial cells to stimulate blood vessel formation in vertebrates [45]. In humans, many viruses seek the up-regulation of VEGF, either by utilizing HIF-1α, COX-2 and AP1 target pathways for the virus-mediated upregulation of VEGF or by the activation of inflammatory mediators [46]. Moreover, VEGFs were found to participate in the immune response of invertebrates [47]. Thus, VGEFs seem to play major role in infection by pathogens, but their role in sea bass is still unclear. On the contrary, the **interferon-induced GTP-binding protein Mx** have a well-characterized antiviral role and show a strict dependence on type-I interferons (IFN α/β) for its expression in different vertebrates. Mx proteins survey exocytic events, target viral nucleocapsid-like structures, and mediate vesicle trafficking to trap essential viral components in order to prevent viral replication at early time points [48]. European sea bass expresses two different Mx genes (MxA and MxB) [49] and the MxA profile in the head kidney during the VNN infection has been studied extensively [11,28,50,51,52]. Different levels of Mx expression have been observed in viral disease-resistant/susceptible fish hosts. An Atlantic salmon family resistant to IPNV demonstrated higher Mx expression in the head kidney compared to susceptible fish, while in Atlantic salmon fry, Mx was moderately up-regulated in resistant fish, but highly up-regulated in susceptible fish at various time points [20]. **Arachidonate 12-lipoxygenase** (alox12) is a lipid mediator which plays a vital role in the innate immune responses of teleost fish following pathogen infection by encoding enzymes that act on different polyunsaturated fatty acid substrates to generate bioactive lipid mediators (e.g., eicosanoids) and mediate inflammatory responses [53,54].

**Calnexin** (canx) is an endoplasmic reticulum (ER) membrane-bound lectin chaperone, which comprises a specialized maturation system combined with the lectin chaperone calreticulin. Its main functions are glycoprotein folding, the quality control of ER protein synthesis, and Ca^2+^ storage, but canx also plays roles in phagocytosis and dendritic cell immunity [55]. Canx is a highly conserved chaperone widely distributed in eukaryotic organisms, and it has been identified in channel catfish [56], zebrafish [57], rainbow trout [58], pufferfish [59], shrimp [55], and crab [60]. Calnexin is highly associated with the correct folding and assembly of MHC molecules in ER [56] and is a potent modulator of antibacterial immune responses [55,59,60]. **Aldehyde dehydrogenase 18A1** (aldh18a1) belongs to the aldehyde dehydrogenase (ALDH) superfamily of enzymes. Human ALDH18A1 encodes a bifunctional enzyme, designated as delta 1-pyrroline-5-carboxylate synthase, which catalyzes the first two steps in proline, ornithine, and arginine biosynthesis [61]. However, its presence and function in teleost fish is not well studied, with the exception of Atlantic salmon and zebrafish [62]. The **DNA damage-inducible transcript 4 protein** (ddit4) is expressed under stress situations, turning off the metabolic activity triggered by the mammalian targeting of rapamycin (mTOR) in humans, and plays a crucial role in cancer [63], but is not well studied in fish. In a recent report on turbot resistance to bacterial infection, ddit4 was expressed at higher levels in the resistant family head kidney and liver, in accordance with our study [21]. Interestingly, in a study on the administration of an aldehyde dehydrogenase (recombinant ALDH7A1) in Atlantic salmon infected with *Aeromonas salmonicida,* it was found that ddit4 was among the differentially expressed transcripts [64].

On the other hand, interactions of the susceptible family are linked by egr1, jun, fosb, mmp13b, mmp19, galr1b, and ptafr. **Early growth response 1** (egr1) is a multifunctional transcription factor capable of both enhancing and/or inhibiting gene expression, highly conserved between numerous species including zebrafish and human. EGR1 can be activated by a wide array of stimuli (e.g., growth factors, cytokines) and various cellular stress states (e.g., viral infections (HSV-1 (herpes simplex 1), HIV (human immunodeficiency virus), EBV (Epstein–Barr virus)), acting either by the activation or suppression of virus infectivity [65]. **Jun** and **Fos** proteins are DNA-binding proteins that are involved in gene expression through transcriptional regulation, including modulating cell proliferation or cell death, in response to different biological stress signals. Both transcription factors are members of the AP1 (activator protein one) complex [66], which in mammals, also mediates gene regulation in response to various stimuli, including cytokines, growth factors, stress signals, and bacterial and viral infections [67].

The matrix metalloproteinase (MMP) gene family is responsible for regulating the degradation of extra cellular matrix (ECM) proteins, which are important for physiological processes, such as wound healing, tissue remodeling, and stress response [68], and they are actively involved in the regulation of the host immune system following infection [69]. The matrix metalloprotease **mmp13** is a collagenase actively involved in the process of bone formation. In teleosts, MMP-13 is required for normal embryogenesis (zebrafish) and is up-regulated following infection (Japanese flounder, channel catfish, and rainbow trout) [17,69,70]. **MMP19** was originally isolated from a rheumatoid arthritis patient. In mammals, MMP19 is able to cleave typical ECM components and the insulin-like growth factor binding protein-3 (IGFBP-3), and it is expressed in macrophages and up-regulated under inflammatory conditions [71]. In teleosts, however, its role is not well understood, with the exception of studies focused on reproduction mechanisms [72]. Interestingly, the **galanin receptor 1b** (galr1b) hub gene is, also, related with the galinergic system, which is mainly involved in reproductive functions and feeding regulation in fish [73].

Overall, the NNV-resistant fish seem to better control their immune responses to the virus compared to the susceptible fish, which appears to have higher inflammatory response following survivors’ re-infection. For example, the same trend has been observed in Atlantic salmon fry infected with IPNV [20] or infectious salmon anemia virus (ISAV) [74], among others. In similar studies, the susceptible fish appeared to have high immune responses, which includes the high representation and expression of inflammatory pathway members, IFN-responsive elements, and cytokines, leading to eventual apoptosis but failing to stop the infection.

## 5. Conclusions

The host immune response to an infectious pathogen is a systematic and complex biological process. The present study offered glimpses of the mechanisms underlying a disseminated systemic response of sea bass belonging to families with varying VVN disease resistance, following survivors’ re-infection. To the best of our knowledge, the present study is the first report of a transcriptome profile comparison of NNV-resistant and -susceptible families of European sea bass following survivors’ re-infection. The obtained transcriptome data showed significant differences between the resistant and susceptible families before and after viral infection. In conclusion, the transcriptome profiles revealed 103 DEGs for the resistant family (RI vs. RNI) and 336 DEGs for the susceptible family (SI vs. SNI). The pathway analysis indicated that immune-related pathways were enriched in both resistant and susceptible fish, but different immune system components were involved in each family response. Apoptosis-related pathways were enriched only in the resistant family, while the ‘necroptosis’ pathway was enriched in both families. Furthermore, protein–protein interaction analysis identified a variety of hub genes for the resistant and the susceptible families, quite distinct in their function on NNV resistance. The presented results offer glimpses of the mechanisms of European sea bass responses to nervous necrosis virus survivors’ re-infection, depending on different host genetic backgrounds, providing valuable datasets for further research on viral disease resistance in teleost fish.

## Figures and Tables

**Figure 1 viruses-17-00230-f001:**
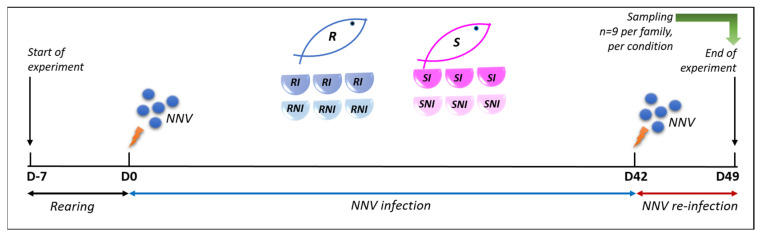
Experimental setup (NNV: nervous necrosis virus; R: resistant; RI: resistant infected; RNI: resistant non-infected; S: susceptible; SI: susceptible infected; SNI: susceptible non-infected; D: days post infection).

**Figure 2 viruses-17-00230-f002:**
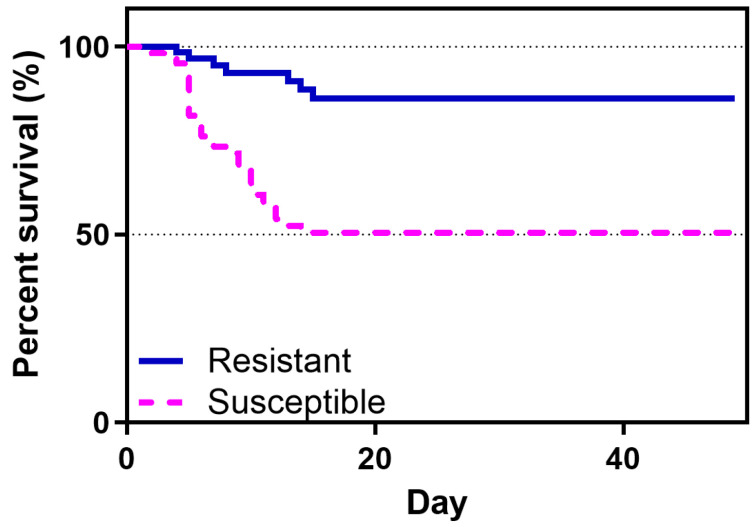
Cumulative survival of NNV-challenged resistant (continuous blue line) and susceptible (dashed magenta line) sea bass families. The mean mortality is represented in each time point (days post infection).

**Figure 3 viruses-17-00230-f003:**
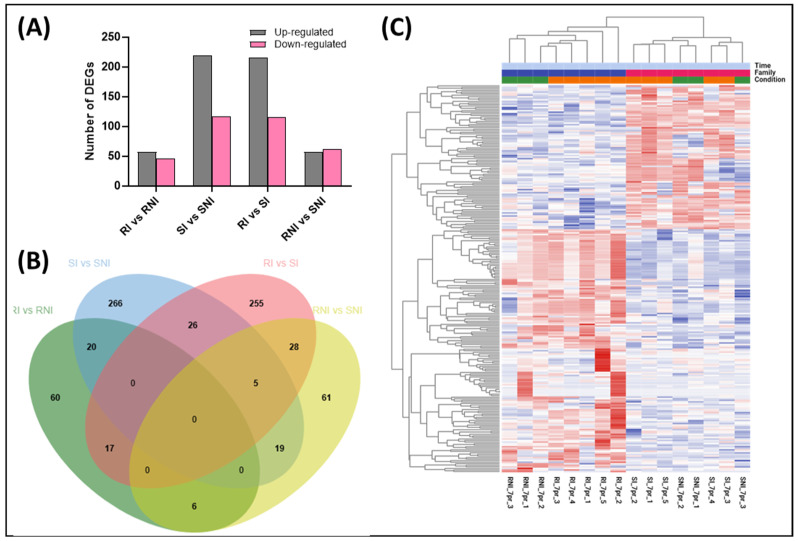
(**A**) Statistics of differentially expressed genes (DEGs) in RI vs. RNI, SI vs. SNI, SI vs. RI, and SNI vs. RNI groups. The X axis and Y axis represent a comparison of samples and the number of DEGs. The first column at each time-point graph corresponds to up-regulated genes, and the second column corresponds to down-regulated genes. (**B**) Venn diagram showing the number of common and exclusive DEGs in RI vs. RNI, SI vs. SNI, SI vs. RI, and SNI vs. RNI groups. (**C**) Cluster analysis of DEGs. Heat map of the genes showed a more than 1.5-fold difference between the resistant (R: blue bar) and susceptible (S: magenta bar) families in NNV-infected (I: orange bar) and non-infected (NI: green bar) experimental groups. Red and blue colors indicate up- and down-regulation in Log2 cpm values, respectively. Each column in the graph represents a sample, each row represents a gene, and the expression of genes in different samples is represented by different colors, with redder colors indicating higher expression and bluer colors indicating lower expression.

**Figure 4 viruses-17-00230-f004:**
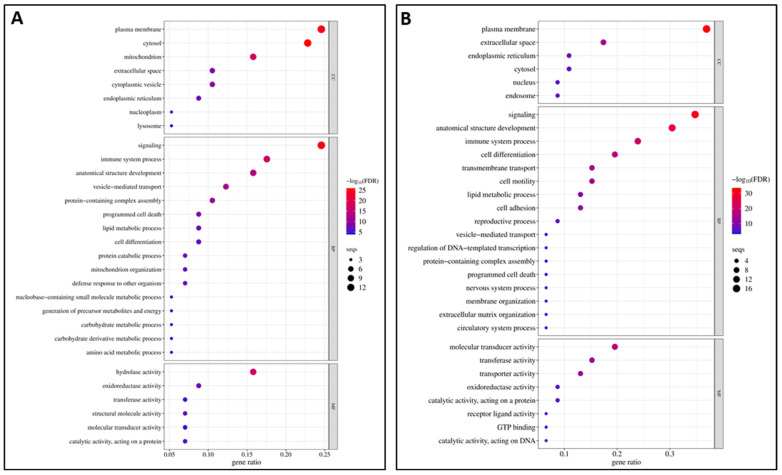
The top 30 gene ontology (GO)-enriched terms in RI vs. RNI groups: (**A**) up-regulated and (**B**) down-regulated DEGs. BP: biological process, MF: molecular function, CC: cellular component.

**Figure 5 viruses-17-00230-f005:**
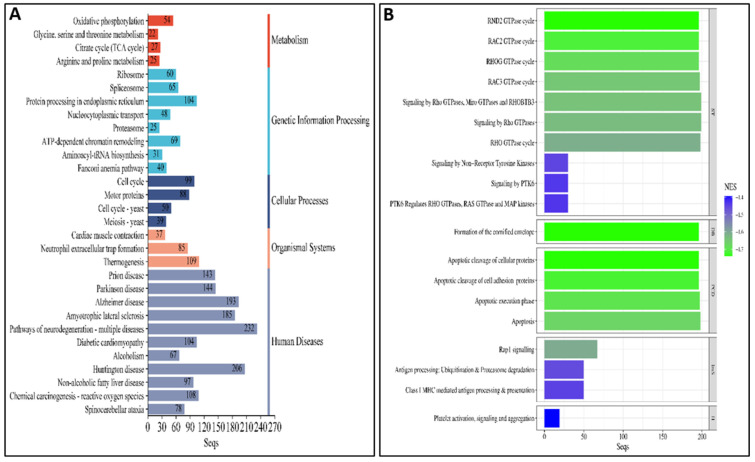
(**A**) The top 30 enriched KEGG pathways in RI vs. RNI groups. (**B**) The enriched Reactome pathways in RI vs. RNI groups. ST: signal transduction; DB: developmental biology; PCD: programmed cell death; ImS: immune system; H: hemostasis; NES: normalized enrichment score.

**Figure 6 viruses-17-00230-f006:**
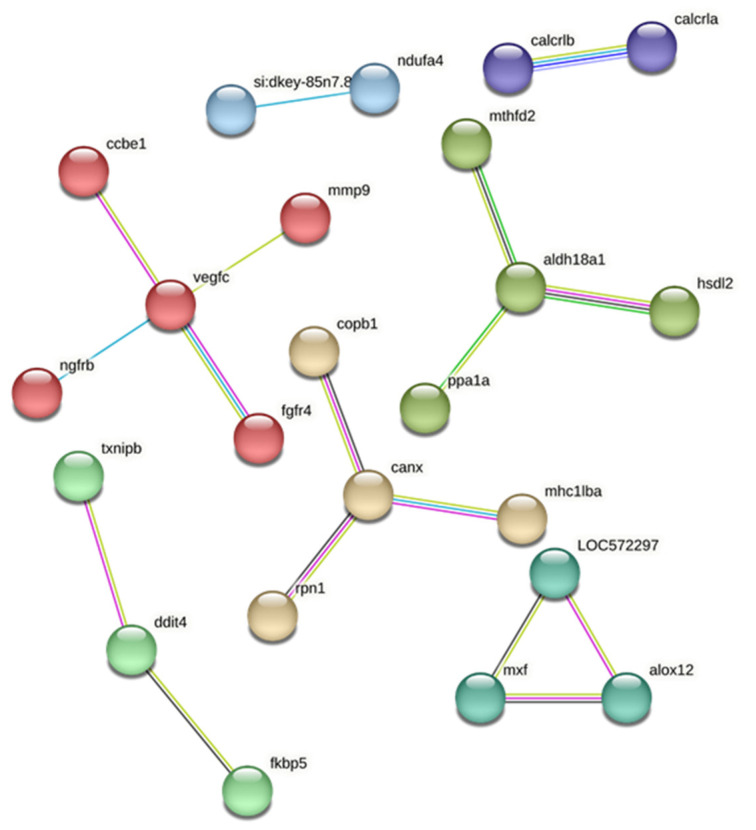
Protein–protein interaction (PPI) networks, with k-means clustering for the zebrafish orthologs of the differentially expressed genes in *D. labrax* RI vs. RNI groups. It is retrieved via API access to the STRING database (https://string-db.org) (accessed on 27 November 2024) and was performed based on the *Danio rerio* protein database. Each colored group represents a different cluster. The edges represent protein–protein interactions.

**Figure 7 viruses-17-00230-f007:**
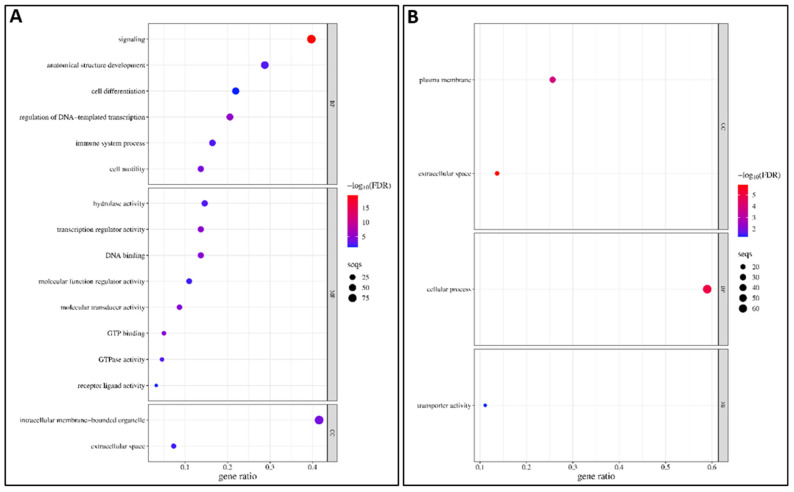
The gene ontology (GO)-enriched terms in SI vs. SNI groups: (**A**) up-regulated and (**B**) down-regulated DEGs. BP: biological process, MF: molecular function, CC: cellular component.

**Figure 8 viruses-17-00230-f008:**
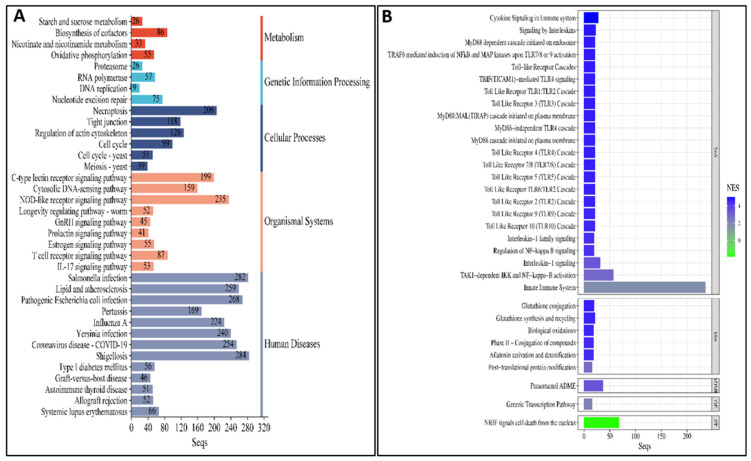
(**A**) The top enriched KEGG pathways in SI vs. SNI groups. (**B**) The enriched Reactome pathways in SI vs. SNI groups. ImS: immune system, Met: metabolism, ADME: drug ADME, GE: gene expression (transcription), ST: signal transduction, NES: normalized enrichment score.

**Figure 9 viruses-17-00230-f009:**
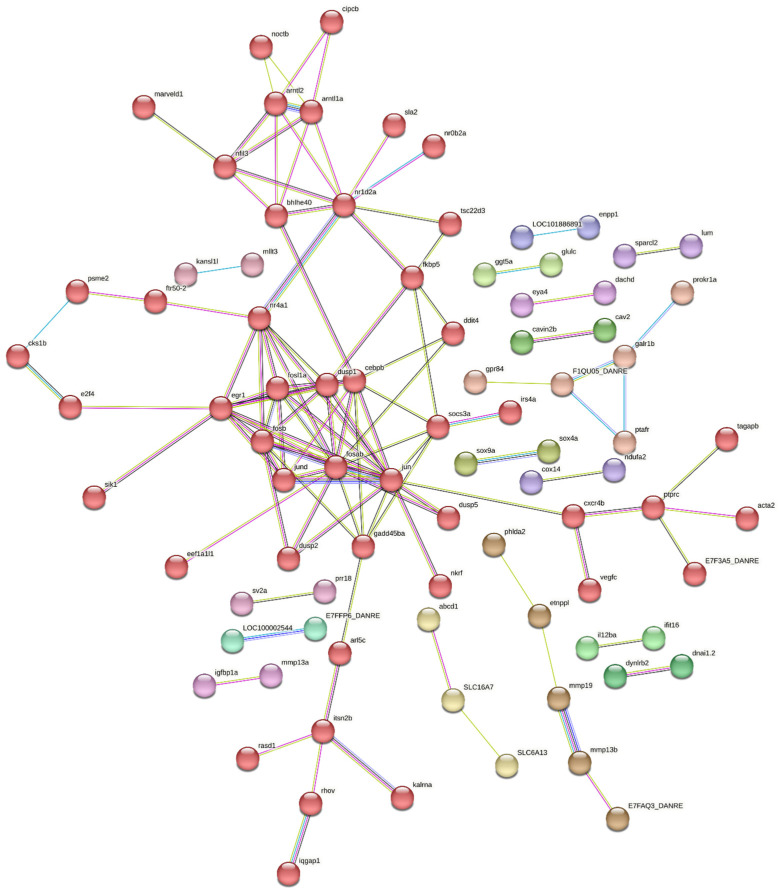
Protein–protein interaction (PPI) networks, with k-means clustering for the zebrafish orthologs of the differentially expressed genes in *D. labrax* SI vs. SNI groups. It is retrieved via API access to the STRING database (https://string-db.org) (accessed on 27 November 2024) and was performed based on the *Danio rerio* protein database. Each colored group represents a different cluster. The edges represent protein-protein interactions.

**Table 1 viruses-17-00230-t001:** Enriched pathway categories on KEGG and Reactome databases for RI vs. RNI groups.

Database	Pathway Category	Found Pathways	Enriched Pathways
KEGG	Human Diseases	96	34
	Genetic Information Processing	31	15
	Metabolism	115	14
	Organismal Systems	91	15
	Cellular Processes	31	10
	Environmental Information Processing	39	9
Reactome	Signal Transduction	141	10
	Programmed Cell Death	13	4
	**Immune System**	131	**3**
	Developmental Biology	30	1
	Hemostasis	26	1

**Table 2 viruses-17-00230-t002:** Enriched pathway categories on KEGG and Reactome databases for SI vs. SNI groups.

Database	Pathway Category	Found Pathways	Enriched Pathways
KEGG	Human diseases	96	13
	Genetic information processing	31	4
	Metabolism	129	4
	Organismal systems	91	9
	Cellular processes	31	6
	Environmental information processing	39	-
Reactome	Immune system	500	23
	Metabolism	2206	5
	Signal transduction	728	4
	Metabolism of proteins	367	1
	Drug ADME	61	1
	Gene expression (Transcription)	252	1

**Table 3 viruses-17-00230-t003:** The ‘uniquely’ implicated components of each family response, resulting from gene ontology (GO), pathway enrichment, and protein–protein (PPI) analysis.

Analysis	Resistant Family		Susceptible Family
**GO enrichment**	vesicle-mediated transport	cytoskeletal protein binding	transcription regulator activity
	transferase activity	transmembrane transport	intracellular membrane-bounded organelle
	cytosol	cell adhesion	GTPase activity
	mitochondrion	endosome	cellular process
	cytoplasmic vesicle	catalytic activity acting on a protein	
	structural molecule activity	nucleus	
	mitochondrion organization	catalytic activity acting on DNA	
	protein catabolic process	circulatory system process	
	catalytic activity acting on a protein	nervous system process	
	carbohydrate metabolic process	extracellular matrix organization	
	amino acid metabolic process	muscle system process	
	generation of precursor metabolites and energy	endocrine process	
	nucleoplasm	cell junction organization	
	nucleobase-containing small molecule metabolic process	inflammatory response	
	chromatin organization	Golgi apparatus	
	nucleocytoplasmic transport	vitamin metabolic process	
	autophagy	extracellular matrix	
	DNA metabolic process	lipid binding	
**Pathway enrichment**	antigen processing and presentation	apoptotic cleavage of cellular and cell adhesion proteins	T cell receptor signaling
	class I MHC-mediated antigen processing and presentation	RNA degradation	cytokine signaling in immune system
	neutrophil extracellular trap formation	mRNA surveillance	toll-like receptor cascades
	cytokine–cytokine receptor interaction	spliceosome	MyD88 dependent cascades
	viral protein interaction with cytokine and cytokine receptor	amino acid metabolism	innate immune system
	cell adhesion molecules	neurodegenerative diseases	interleukin-1 family signaling
	Rap1 signaling		regulation of NF-kappa B signaling
	HIF-1 signaling		NRIF signals cell death from the nucleus
	Ras signaling		cell cycle
	signaling by PTK6, RHO GTPase cycle		regulation of actin cytoskeleton
	apoptosis		nucleotide excision repair
	p53 signaling		nicotinate and nicotinamide metabolism
**PPI**	vegfc		egr1
	aldh18a1		jun
	canx		fosb
	mxf		mmp13b and mmp19
	alox12		galr1b
	ddit4		

## Data Availability

The raw sequencing data are available in the NCBI database (accession no. PRJNA1030357).

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
