# Peer review of "Influence of Viral Re-Infection on Head Kidney Transcriptome of Nervous Necrosis Virus-Resistant and -Susceptible European Sea Bass (Dicentrarchus labrax, L.)"

_viruses, 2025, doi:10.3390/v17020230_

Round 1
Reviewer 1 Report
Comments and Suggestions for Authors
In the present study, in order to investigate the VNN resistance mechanism of European sea bass, two genetically distinct families were chosen for head kidney transcriptome analysis. Based on the transcriptome data, the resistant and susceptible families’ responses were assessed in three complementary ways, i.e. gene ontology (GO) enrichement analysis, pathway (KEGG and reactome) analysis and protein-protein interaction (PPI) analysis, which were used to reveal key genes and pathways important for disease resistance. In general, this is a interesting topic.
1. Line 118-120, A total of 300 randomly selected fish from both families (weight: 125.85 ± 29.8 g) were acclimatized for 3-4 days in 1 m3 cylindroconical fiberglass tanks connected to a closed recirculated sea water system, with 20 m3 total volume capacity. Can the fish used be infected with NNV? Fish are larger, and NNV is more likely to infect smaller fish.
2. The target organ for NNV infection is the brain, so why was the head kidney chosen for transcriptomic analysis?
Reviewer 2 Report
Comments and Suggestions for Authors
The paper “Influence of Viral Re-Infection on Head Kidney Transcriptome of Nervous Necrosis Virus Resistant and Susceptible European Sea Bass (Dicentrarchus labrax, L.)” study how NNV re-infection affects the European sea bass (Dicentrarchus labrax, L.) transcriptome, in disease resistant and susceptible sea bass families.
Some comment are stated below:
1. Why perform re-infection? In my opinion, this study is not an infection analysis, it’s a study on the attenuated vaccine.
2. What is the origin of SSN-1 cell? European Sea Bass or not? If it is not European Sea Bass, the control fish should be injected with the cellular supernatant.
3. Section 2.1 and 2.2, the description of the method is disorganized, this two section should be rewrote.
4. Section 3.5, 6 genes is not enough.
